# Identifying images in the biology literature that are problematic for people with a color-vision deficiency

Harlan P Stevens, Carly V Winegar, Arwen F Oakley, Stephen R Piccolo*

Department of Biology, Brigham Young University, Provo, United States

**Abstract** To help maximize the impact of scientific journal articles, authors must ensure that article figures are accessible to people with color-vision deficiencies (CVDs), which affect up to 8% of males and 0.5% of females. We evaluated images published in biology- and medicine-oriented research articles between 2012 and 2022. Most included at least one color contrast that could be problematic for people with deuteranopia ('deuteranopes'), the most common form of CVD. However, spatial distances and within-image labels frequently mitigated potential problems. Initially, we reviewed 4964 images from *eLife*, comparing each against a simulated version that approximated how it might appear to deuteranopes. We identified 636 (12.8%) images that we determined would be difficult for deuteranopes to interpret. Our findings suggest that the frequency of this problem has decreased over time and that articles from cell-oriented disciplines were most often problematic. We used machine learning to automate the identification of problematic images. For a hold-out test set from *eLife* (n=879), a convolutional neural network classified the images with an area under the precision-recall curve of 0.75. The same network classified images from PubMed Central (n=1191) with an area under the precision-recall curve of 0.39. We created a Web application (https://bioapps.byu.edu/colorblind_image_tester); users can upload images, view simulated versions, and obtain predictions. Our findings shed new light on the frequency and nature of scientific images that may be problematic for deuteranopes and motivate additional efforts to increase accessibility.

**\*For correspondence:**
stephen_piccolo@byu.edu

**Competing interest:** The authors declare that no competing interests exist.

## eLife assessment

In this **important** study, the authors manually assessed randomly selected images published in eLife between 2012 and 2022 to determine whether they were accessible for readers with deuteranopia, the most common form of color vision deficiency. They then developed an automated tool designed to classify figures and images as either "friendly" or "unfriendly" for people with deuteranopia. Such a tool could be used by journals or researchers to monitor the accessibility of figures and images, and the evidence for its utility was **solid**: it performed well for eLife articles, but performance was weaker for a broader dataset of PubMed articles, which were not included in the training data. The authors also provide code that readers can download and run to test their own images, and this may be of most use for testing the tool, as there are already several free, user-friendly recoloring programs that allow users to see how images would look to a person with different forms of color vision deficiency. Automated classifications are of most use for assessing many images, when the user does not have the time or resources to assess each image individually.

## Introduction

Most humans have trichromatic vision: they perceive blue, green, and red colors using three types of retinal photoreceptor cells that are sensitive to short, medium, or long wavelengths of light, respectively. Color-vision deficiency (CVD) affects between 2% and 8% of males (depending on ancestry) and approximately 0.5% of females (*Delpero et al., 2005*). Congenital CVD is commonly caused by mutations in the genes (or nearby promoter regions) that code for red or green cone photopigments; these genes are proximal to each other on the X chromosome (*Nathans et al., 1986*).

CVD is divided into categories, the most common being *deutan CVD*, affecting approximately 6% of males of European descent, and *protan CVD*, affecting 2% of males of European descent (*Delpero et al., 2005*). Both categories are commonly known as red-green colorblindness. Within each category, CVD is subclassified according to whether individuals are *dichromats*—able to see two primary colors—or *anomalous trichromats*—able to see three primary colors but differently from *normal trichromats*. Anomalous trichromats differ in the degree of severity with which they can distinguish color patterns. Individuals with deuteranopia ('deuteranopes') or protanopia do not have corresponding green or red cones, respectively (*Simunovic, 2010*). Individuals with deuteranomaly do not have properly functioning green cones, and those with protanomaly do not have properly functioning red cones. People with any of these conditions often see green and red as brown or beige colors. Thus, when images contain shades of green and red—or when either is paired with brown—parts of the image may be indistinguishable. Furthermore, it can be problematic when some pinks or oranges are paired with greens. These issues can lead to incorrect interpretations of figures in scientific journal articles for individuals with CVD.

Efforts have been made to ensure that scientific figures are accessible to people with CVD. For example, researchers have developed algorithms that attempt to recolor images so that people with CVD can more easily interpret them (*Flatla, 2011*; *Lin et al., 2019*; *Tsekouras et al., 2021*). However, these tools are not in wide use, and more work is needed to verify their efficacy in practice. In the meantime, as researchers prepare scientific figures, they can take measures to improve accessibility for people with CVD. For example, they can avoid rainbow color maps that show colors in a gradient; they can use color schemes or color intensities that are CVD friendly (*Crameri et al., 2020*); additionally, they can provide labels that complement information implied by color differences. However, for the millions of images that have already been published in scientific articles, little is known about the frequency with which these images are CVD friendly. The presence or absence of particular color pairings—and distances between them—can be quantified computationally to estimate this frequency. However, a subjective evaluation of individual images is necessary to identify whether color pairings and distances are likely to affect scientific interpretation.

In this paper, we focus on deuteranopia and its subtypes. To estimate the extent to which the biological and medical literature contains images that may be problematic to deuteranopes, we manually reviewed a 'training set' of 4964 images and two 'test sets' of 879 and 1191 images, respectively. These images were published in articles between the years 2012 and 2022. After identifying images that we deemed most likely to be problematic or not, we used machine-learning algorithms to identify patterns that could discriminate between these two categories of images and thus might be useful for automating the identification of problematic images. If successful, such an algorithm could be used to alert authors, presenters, and publishers that scientific images could be modified to improve visual accessibility and thus make biological and medical fields more inclusive.

## Results

We downloaded images from research articles published in the *eLife* journal. Not counting duplicate versions of the same image, we obtained 66,253 images. Of these images, 1744 (2.6%) were grayscale (no color). Of these images, 56,816 (85.6%) included at least one color pair for which the amount of contrast might be problematic to people with moderate-to-severe deuteranopia ('deuteranopes'). To characterize potentially problematic aspects of each color-based image, we calculated five metrics based on color contrasts and distances; we also compared the color profiles against what deuteranopes might see. The *mean, pixel-wise color distance between the original and simulated image* exhibited a bimodal distribution, according to Hartigans' Dip Test for Unimodality (p<0.001; *Hartigan and Hartigan, 1985*). Specifically, 4708 images (7.3%) had a difference smaller than 0.01,

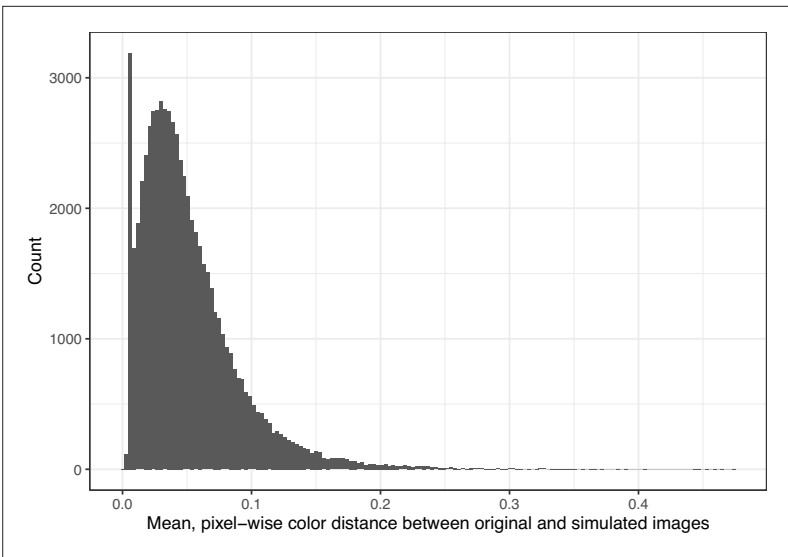

**Figure 1.** Mean, pixel-wise color distance between each original and simulated image from *eLife*. The histogram depicts the frequency distribution of this metric for 64,509 non-grayscale images.

while the median difference for the remaining images was 0.05 (*Figure 1*). Most other metrics showed similar patterns, although bimodality was less apparent through visual inspection (*Figure 2*; *Figure 3*; *Figure 4*). The exception was the *proportion of pixels in the original image that used a color from one of the high-ratio color pairs*, which was unimodal (p=1; *Figure 5*).

We determined that many images with the highest (or lowest, as would be the case for the 'Mean Euclidean distance between pixels for high-ration color pairs') scores for these metrics would be problematic for deuteranopes. However, we noted that certain color pairs were more problematic than others, and the use of effective labels and/or spacing between colors often mitigated potential problems. Thus, to better estimate the extent to which images are problematic for deuteranopes, we manually reviewed a sample of 4964 images and judged whether it would be likely for deuteranopes to recognize the scientific message behind each image. *Supplementary file 2* contains a record of these evaluations, along with comments that indicate either problematic aspects of the images or factors that mitigated potential problems. We concluded that 636 (12.8%) of the images were

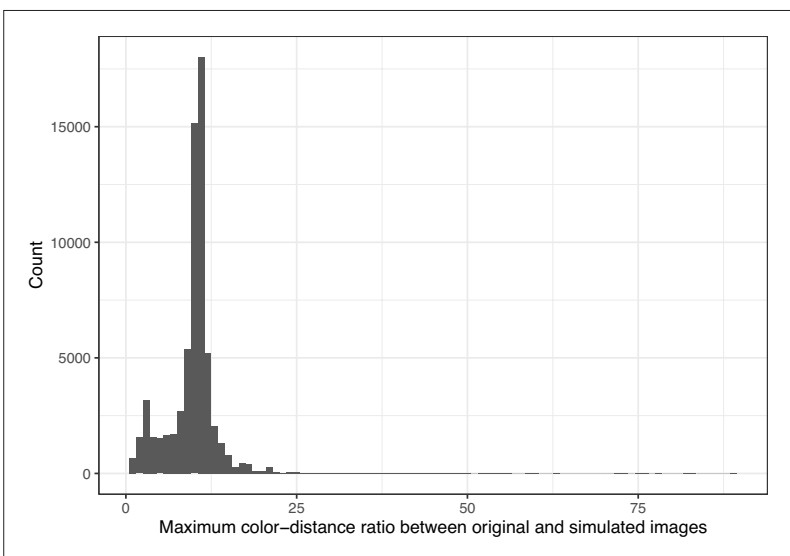

**Figure 2.** Maximum color-distance ratio between each original and simulated image from *eLife*. The histogram depicts the frequency distribution of this metric for 64,509 non-grayscale images.

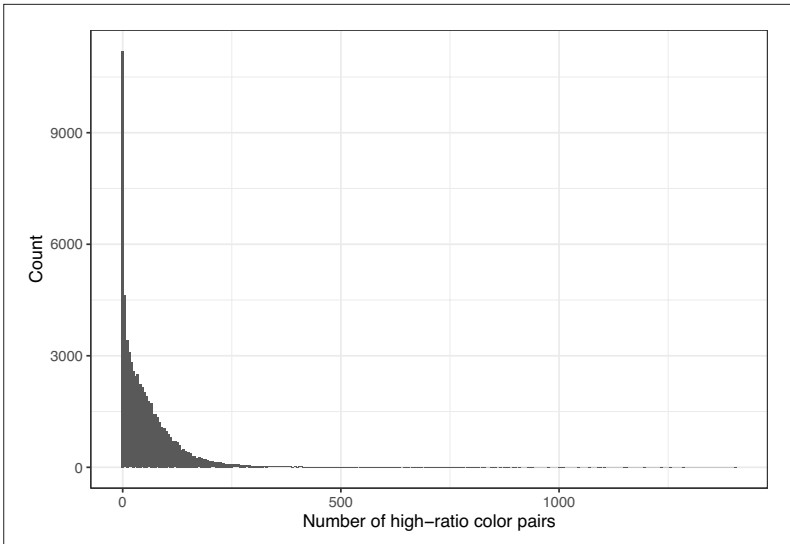

**Figure 3.** Number of color pairs per image that exhibited a high color-distance ratio between the original and simulated images from *eLife*. The histogram depicts the frequency distribution of this metric for 64,509 non-grayscale images.

'Definitely problematic', whereas 3865 of the images (77.9%) were 'Definitely okay'. The remaining images were grayscale (n=179), or we were unable to reach a confident conclusion (n=284). For the images that were 'Definitely okay', we visually detected shades of green and red or orange in 2348 (60.8%) images; however, in nearly all (99.3%) of these cases, we deemed that the contrasts between the shades were sufficient that a deuteranope could interpret the images. Furthermore, distance between the colors and/or labels within the images mitigated potential problems in 54.2% and 48.4% of cases, respectively.

We evaluated longitudinal trends and differences among biology subdisciplines. In some cases for the *eLife* articles, multiple images came from the same journal article. Therefore, to avoid pseudoreplication, we categorized each *article* as either 'Definitely okay' or 'Definitely problematic'. If an article included at least one 'Definitely problematic' image, we categorized the entire article under this category. The percentage of 'Definitely problematic' articles declined steadily between 2012 and 2021,

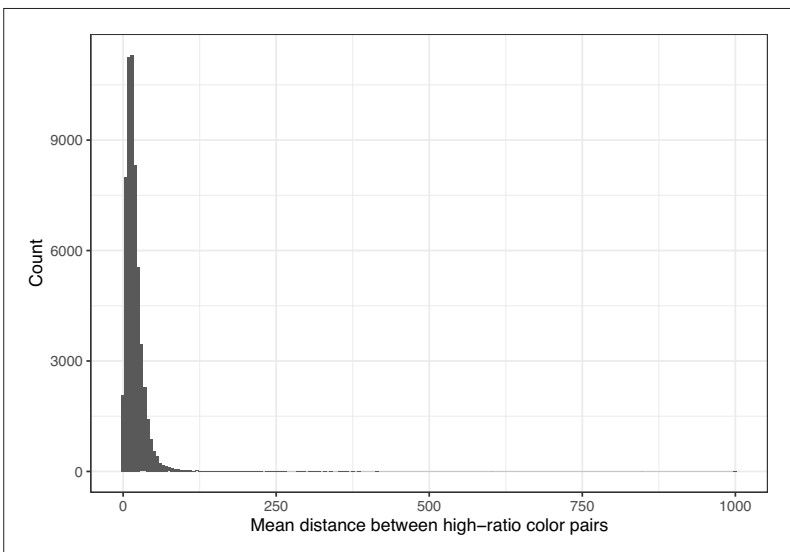

**Figure 4.** Mean Euclidean spatial distance per image between pixels for high-ratio color pairs from *eLife*. The histogram depicts the frequency distribution of this metric for 64,509 non-grayscale images.

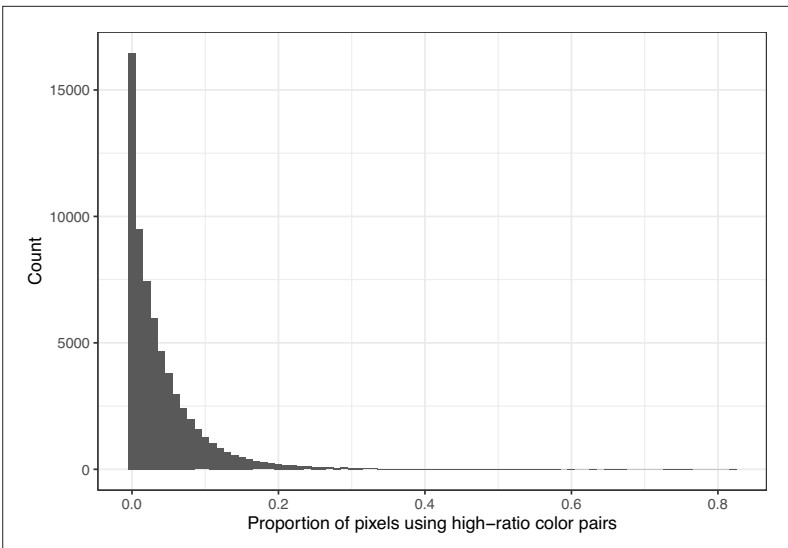

**Figure 5.** Proportion of pixels in each original image that used a color from one of the high-ratio color pairs from *eLife*. The histogram depicts the frequency distribution of this metric for 64,509 non-grayscale images.

with a modest increase in 2022 (***Figure 6***). (Fewer articles were available for 2022 than for prior years.) Using a generalized linear model with a binomial family to perform logistic regression, we found this decline to be statistically significant (p<0.001). A $\chi^2$ goodness-of-fit test revealed that the number of 'Definitely problematic' articles differed significantly by subdiscipline (p<0.001). The subdisciplines with the highest percentages of problematic articles were *Cell Biology*, *Developmental Biology*, and *Stem Cells and Regenerative Medicine* (***Figure 7***). The subdisciplines with the lowest percentages of problematic articles were *Evolutionary Biology*, *Genetics and Genomics*, and *Computational and Systems Biology*.

Despite the benefits of manual review, this process is infeasible on a large scale. Therefore, we evaluated techniques for automating image classification. As an initial test, we used the five image-quantification metrics. We also combined these into a single, ranked-based score for each image. In all cases, the metrics differed significantly between the 'Definitely okay' and 'Definitely problematic' images (***Figure 8***; ***Figure 9***; ***Figure 10***; ***Figure 11***; ***Figure 12***; ***Figure 13***). To estimate their predictive performance, we performed cross validation on the training set. Values relatively close to 1.0 indicate

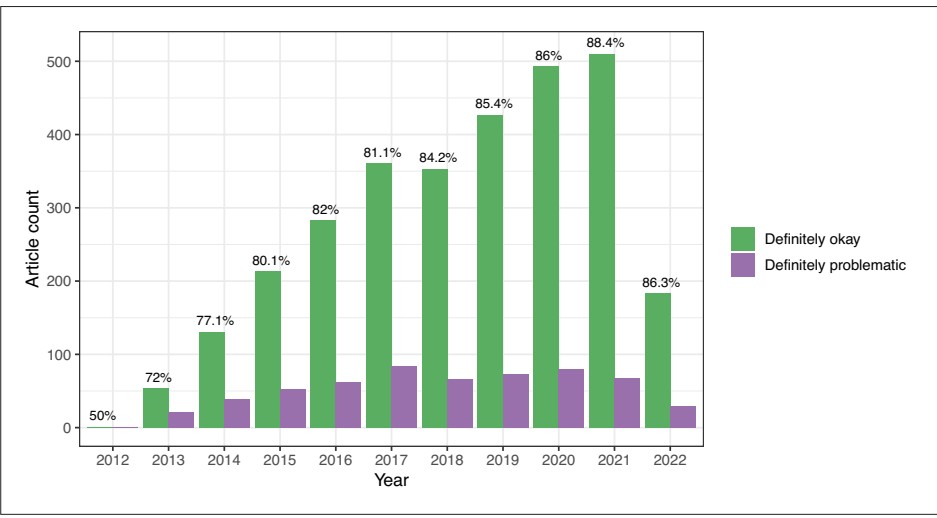

**Figure 6.** Longitudinal trends for the *eLife* articles. For the training set, we summarized our findings per article. This graph shows article counts for the 'Definitely okay' and 'Definitely problematic' categories for each year evaluated.

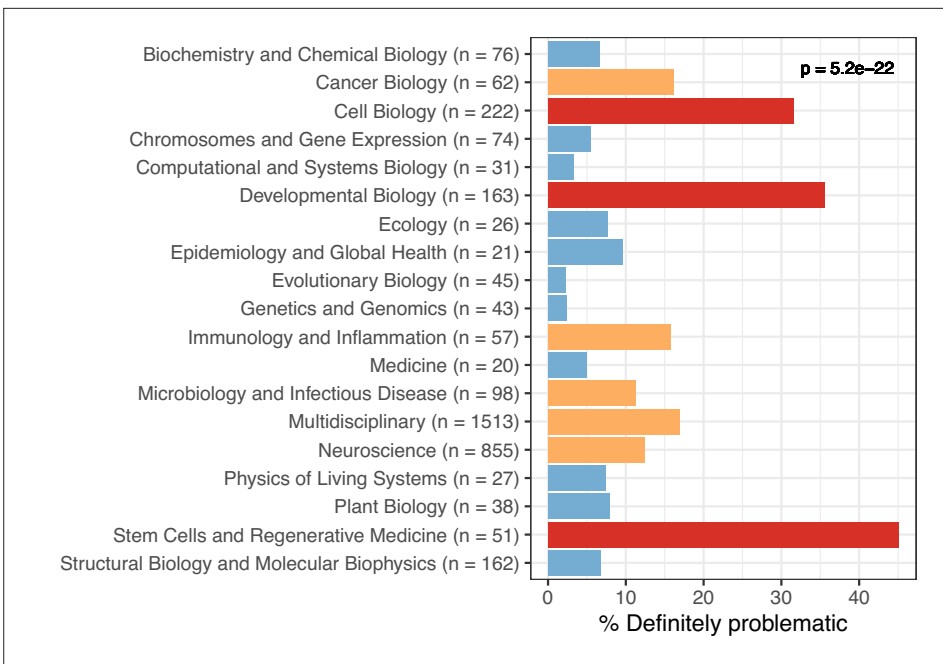

**Figure 7.** Trends by biology subdiscipline for the *eLife* articles. For the training set, this graph shows the percentage of articles categorized as 'Definitely problematic' for a given subdiscipline, as indicated in the article metadata. In many cases, a single article was associated with multiple subdisciplines; these articles are shown as 'Multidisciplinary'. We used a $\chi^2$ goodness-of-fit test to calculate the p-value, with the overall proportion of each discipline as the expected probability.

relatively high performance. A value of 0.5 indicates that predictions are no better than random guessing. The best-performing metric was *number of color pairs that exhibited a high color-distance ratio between the original and simulated images* (AUROC: 0.75; AUPRC: 0.34). All the other metrics— except *mean, pixel-wise color distance between the original and simulated image*—performed better than random guessing (*Supplementary file 1A*). As an alternative to the combined rank score, we used classification algorithms to make predictions with the five metrics as inputs. In cross-validation on

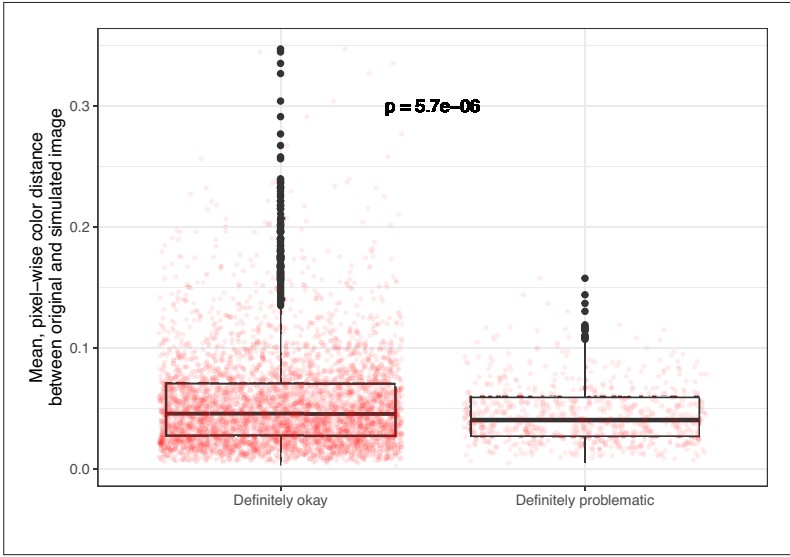

**Figure 8.** Mean, pixel-wise color distance between each original and simulated image from *eLife* categorized as 'Definitely okay' or 'Definitely problematic'. We used a two-sided Mann-Whitney U test to calculate the p-value.

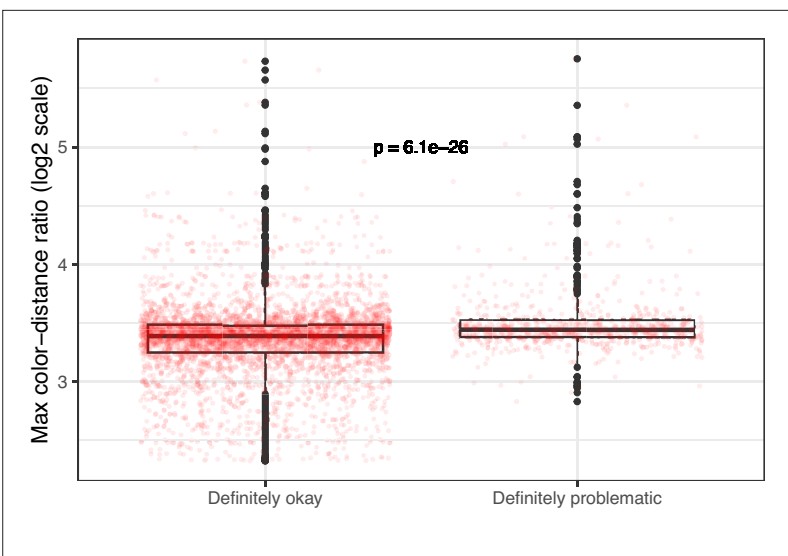

**Figure 9.** Maximum color-distance ratio between each original and simulated image from *eLife* categorized as 'Definitely okay' or 'Definitely problematic'. We used a two-sided Mann-Whitney U test to calculate the p-value.

the training set, the best-performing algorithm was Logistic Regression (AUROC: 0.82; AUPRC: 0.43; *Supplementary file 1B*).

Additionally, we created a convolutional neural network (CNN) to make predictions according to visual and spatial patterns in the images. CNNs are highly configurable and often sensitive to model parameters and configurations. Accordingly, we performed multiple iterations of cross validation on the training set and compared a variety of hyperparameters and configurations. All non-default options performed better than Logistic Regression based on the image-quantification metrics (*Supplementary file 1C*). The best-performing approach used class weighting; early stopping; random flipping and rotation (threshold: 0.2); a dropout rate of 0.5; the ResNet pre-trained model for transfer learning; and model fine tuning.

We manually reviewed a hold-out test set that consisted of 1,000 additional images from *eLife* (*Supplementary file 3*). After we removed images that were not 'Definitely okay' or 'Definitely problematic', 879 images remained. For the Logistic Regression algorithm and CNN, we trained models

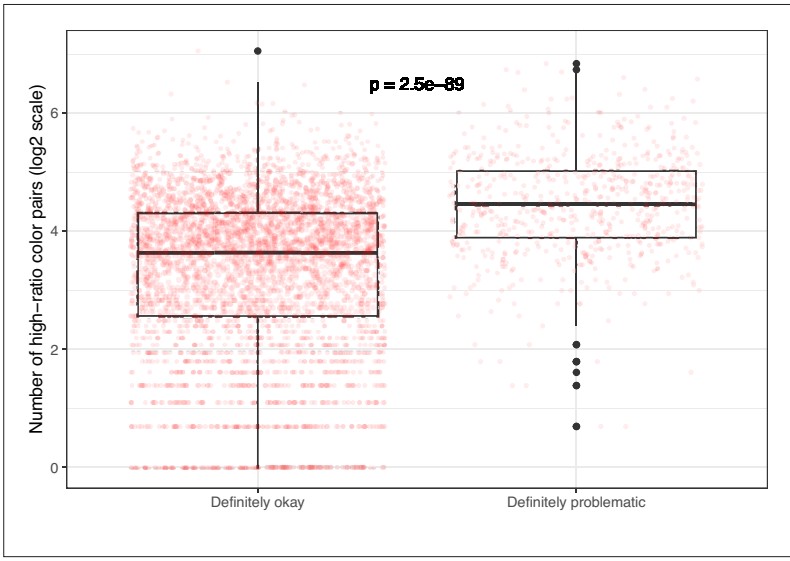

**Figure 10.** Number of high-ratio color pairs per image from *eLife* categorized as 'Definitely okay' or 'Definitely problematic'. We used a two-sided Mann-Whitney U test to calculate the p-value.

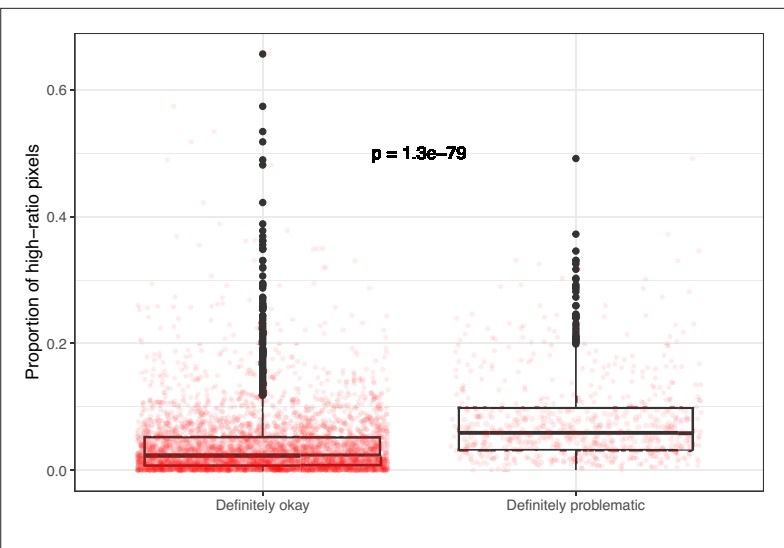

**Figure 11.** Proportion of pixels for high-ratio color pairs for images from *eLife* categorized as 'Definitely okay' or 'Definitely problematic'. We used a two-sided Mann-Whitney U test to calculate the p-value.

using the full training set and classified each hold-out image as 'Definitely okay' or 'Definitely problematic'. Logistic Regression classified the images with an AUROC of 0.82 (AUPRC: 0.49; *Figure 14*; *Figure 15*; *Figure 16*). The CNN classified the images with an AUROC of 0.89 (AUPRC: 0.77; *Figure 17*; *Figure 18*; *Figure 19*). *Supplementary file 4* indicates the performance of both models for a variety of classification metrics.

For the 92 hold-out images that were misclassified by the CNN model, we compared them against our manual annotations and determined that in 13 cases, the reviewers had missed subtle patterns; we conclude that it would be justified to change these labels (*Supplementary file 5*). For 31 of the misclassified images, we visually identified patterns that might have confused the CNN; however, upon reevaluation, we maintain that the original labels were valid. For the remaining 48 misclassified images, we were unable to identify patterns that seemed likely to have confused the model.

Lastly, we used the Logistic Regression and CNN models to predict 'Definitely okay' or 'Definitely problematic' status for a second hold-out set with images from PubMed Central (*Supplementary file*

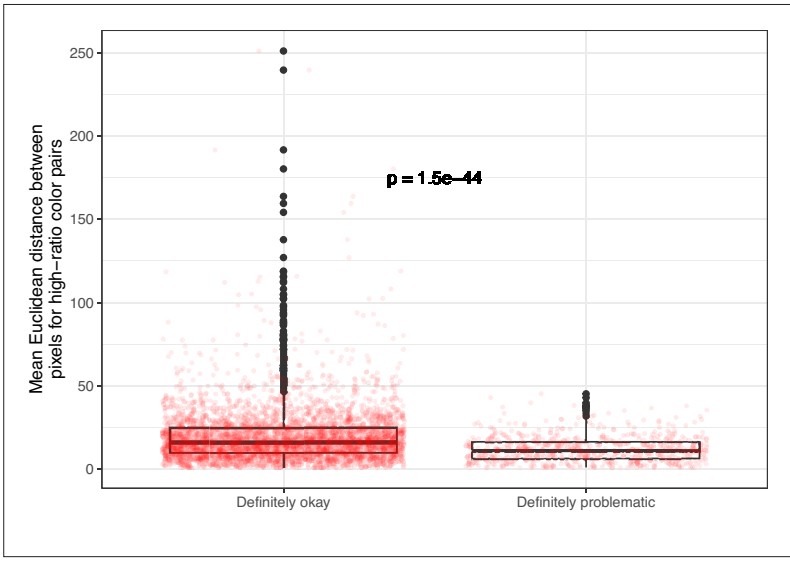

**Figure 12.** Mean, pixel-wise Euclidean distance for high-ratio color pairs in images from *eLife* categorized as 'Definitely okay' or 'Definitely problematic'. We used a two-sided Mann-Whitney U test to calculate the p-value.

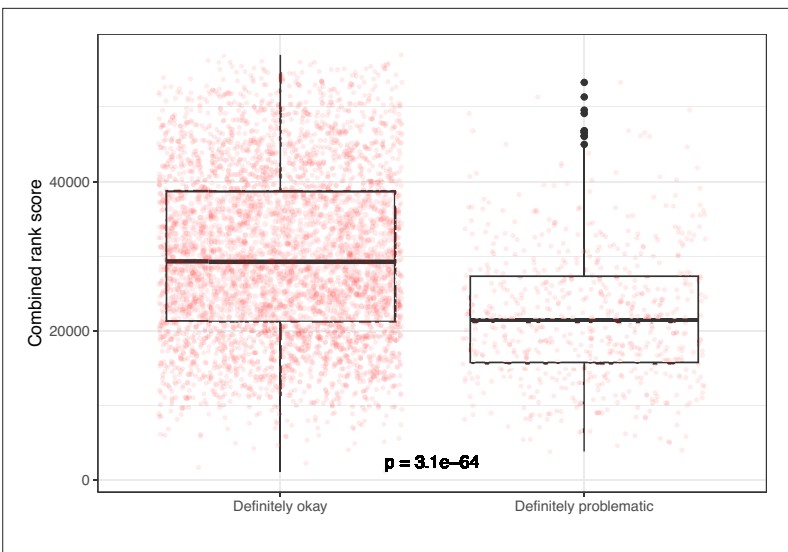

**Figure 13.** Rank-based metric score for images from *eLife* categorized as 'Definitely okay' or 'Definitely problematic'. We used a two-sided Mann-Whitney U test to calculate the p-value.

*6*). Of the non-grayscale images, 104 were labeled as 'Definitely problematic' and 1087 as 'Definitely okay'. Again, the CNN model (AUROC: 0.78; AUPRC: 0.39; *Figure 20*; *Figure 21*; *Figure 22*) outper-formed the Logistic Regression model (AUROC: 0.73; AUPRC: 0.16; *Figure 23*; *Figure 24*; *Figure 25*; *Supplementary file 4*).

## Discussion

To learn more about the prevalence of images in the biology literature that may be inaccessible to deuteranopes, we examined thousands of articles from the *eLife* journal. *eLife* uses a content-licensing scheme that made it possible to perform this study in a transparent manner. Additionally, we

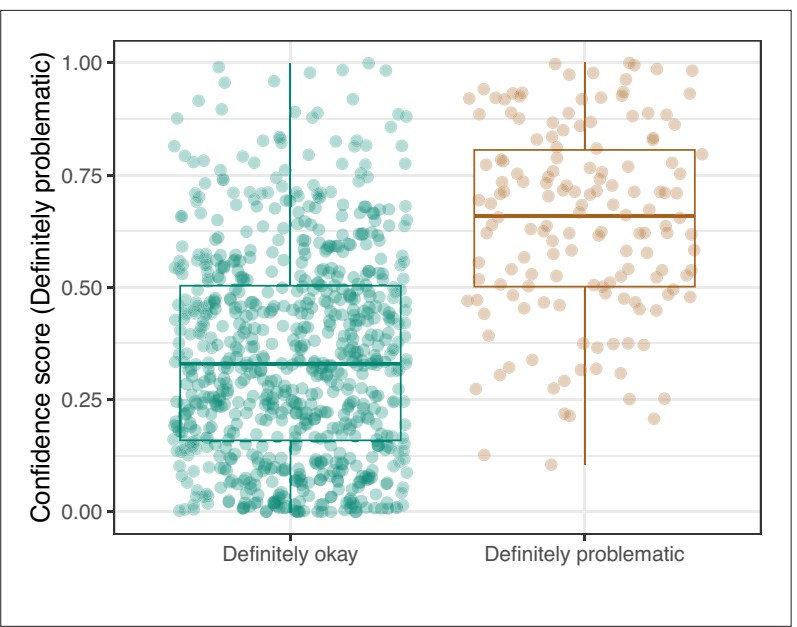

**Figure 14.** Logistic Regression predictions for the images in the *eLife* hold-out test set. Each point represents the prediction for an individual image. Relatively high confidence scores indicate that the model had more confidence that a given image was 'Definitely problematic' for a person with deuteranopia.

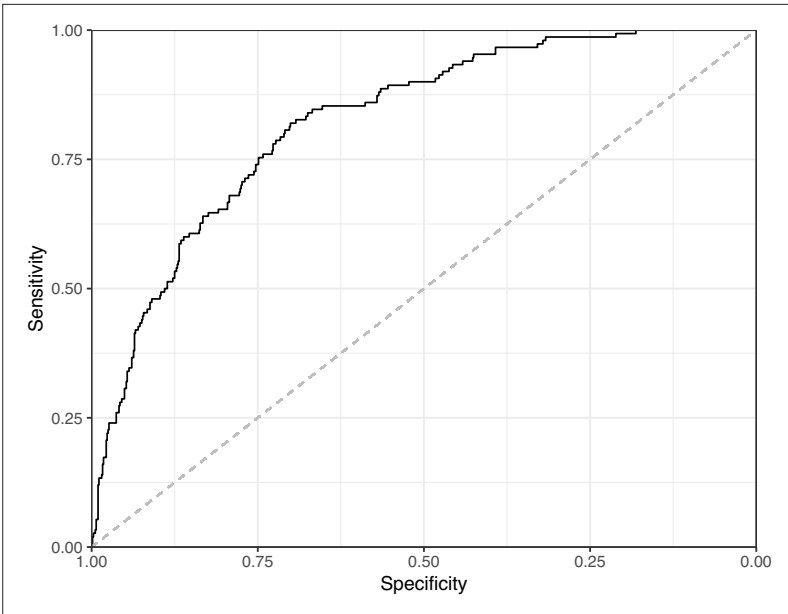

**Figure 15.** Receiver operating characteristic curve for the Logistic Regression predictions for the images in the *eLife* hold-out test set. This curve illustrates tradeoffs between sensitivity and specificity. The area under the curve is 0.82. The dashed, gray line indicates the performance expected by random chance.

selected articles from PubMed Central, as images from these articles represent life-science journals more broadly. After manual review, we estimate that 12.8% of the figures in *eLife* would be challenging to interpret for scientists with moderate-to-severe deuteranopia. The percentage of 'Definitely problematic' figures in PubMed Central articles was considerably lower (5.2%). One reason is that a much higher percentage (38.5%) of the images from PubMed Central were grayscale compared to those from *eLife* (4.4%). The findings for both sources indicate that color accessibility is a problem for thousands of journal articles per year.

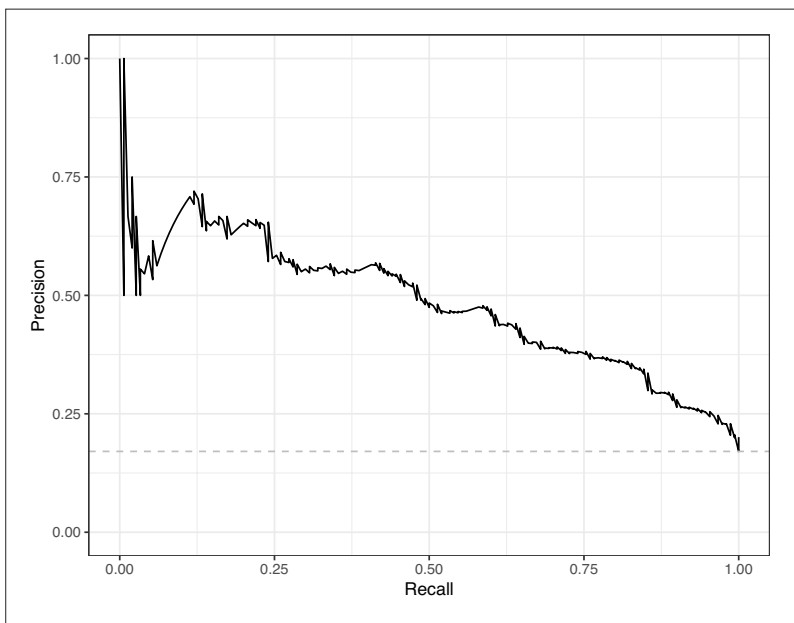

**Figure 16.** Precision-recall curve for the Logistic Regression predictions for the images in the *eLife* hold-out test set. This curve illustrates tradeoffs between precision and recall. The area under the curve is 0.49. The dashed, gray line indicates the frequency of the minority class ('Definitely problematic' images).

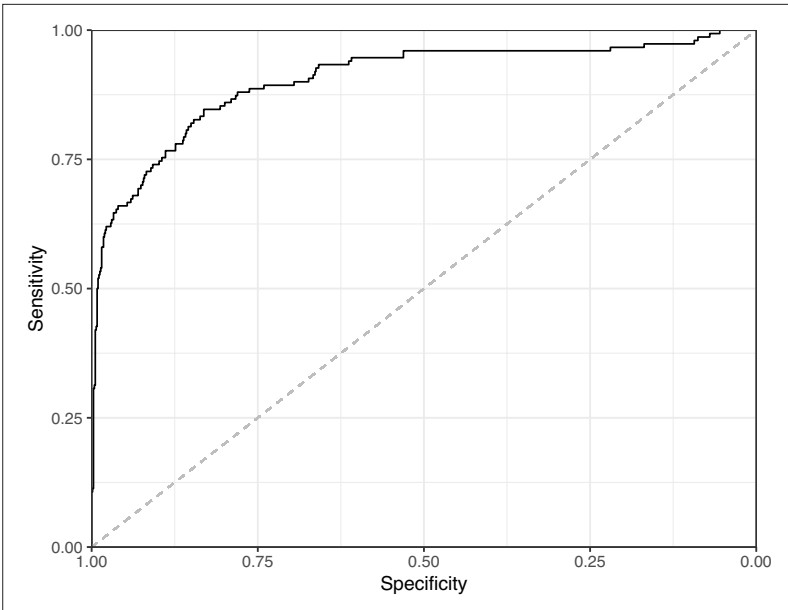

**Figure 17.** Receiver operating characteristic curve for the Convolutional Neural Network predictions for the images in the *eLife* hold-out test set. This curve illustrates tradeoffs between sensitivity and specificity. The area under the curve is 0.89. The dashed, gray line indicates the performance expected by random chance.

Significant work has been done to address and improve accessibility for individuals with CVD (*Zhu and Mao, 2021*). This work can be categorized into four types of studies: simulation methods, recolorization methods, estimating the frequency of accessible images, and educational. Simulation methods have been developed to better understand how images appear to individuals with CVD. Brettel et al. first simulated CVDs using the long, medium, and short (LMS) colorspace (*Brettel et al., 1997*). For dichromacy, the colors in the LMS space are projected onto an axis that corresponds to the non-functional cone cell. Viénot et al. expanded on this work by applying a 3x3 transformation matrix

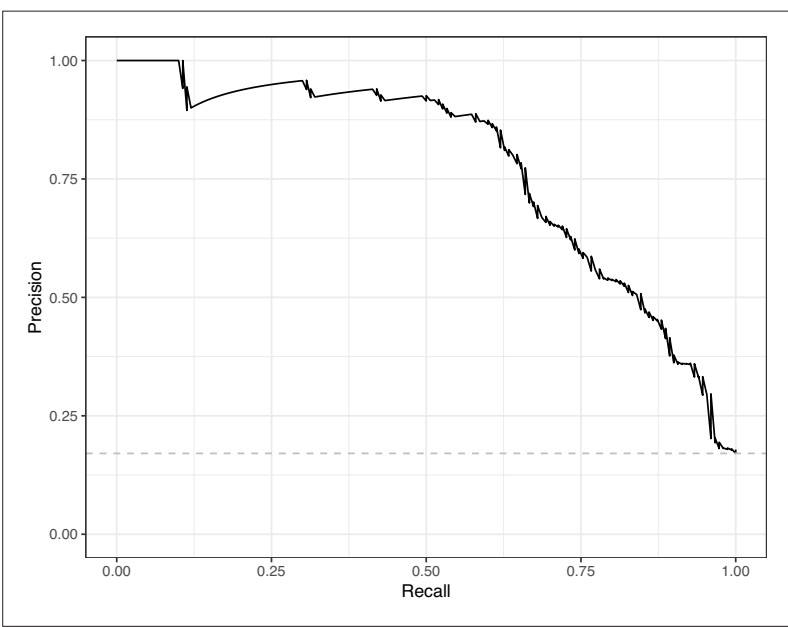

**Figure 18.** Precision-recall curve for the Convolutional Neural Network predictions for the images in the *eLife* hold-out test set. This curve illustrates tradeoffs between precision and recall. The area under the curve is 0.75. The dashed, gray line indicates the frequency of the minority class ('Definitely problematic' images).

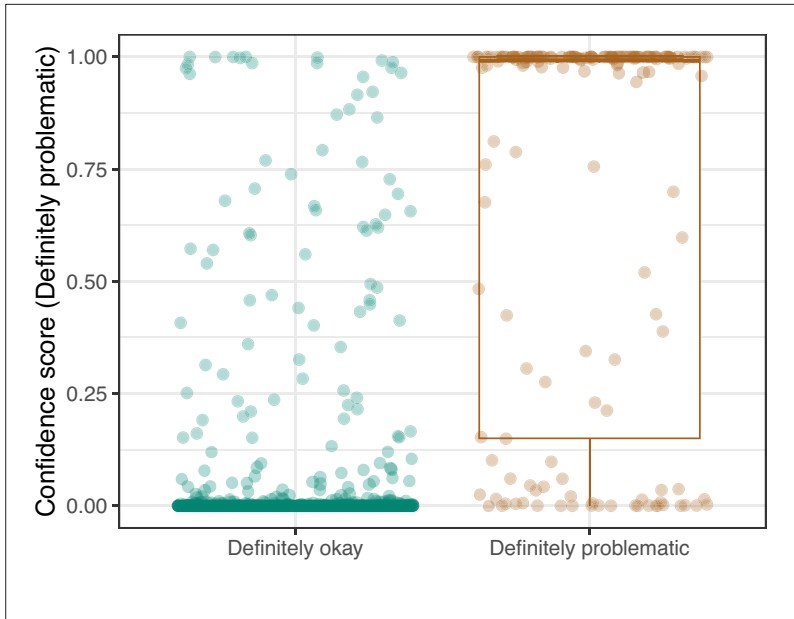

**Figure 19.** Convolutional Neural Network predictions for images in the *eLife* hold-out test set. Each point represents the prediction for an individual image. Relatively high confidence scores indicate that the model had more confidence that a given image was 'Definitely problematic' for a person with deuteranopia.

to simulate images in the same LMS space (*Vinot et al., 1999*). Machado et al. created matrices to simulate CVDs based on the shift theory of cone cell sensitivity (*Machado et al., 2009*; *Stockman and Sharpe, 2000*). These algorithms allow individuals without CVD to qualitatively test how their images might appear to people with CVD. The simulation algorithms and matrices are freely available and accessible via websites and software packages (*Coblis, 2021*; *Color blind, 2020*; *DaltonLens, 2023*; *Wilke, 2023*).

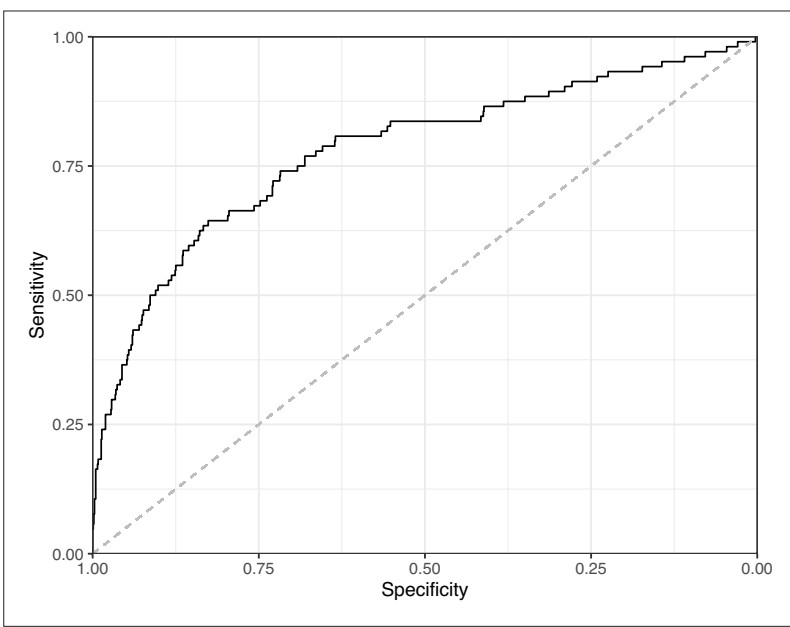

**Figure 20.** Receiver operating characteristic curve for the Convolutional Neural Network predictions for the images in the PubMed Central hold-out test set. This curve illustrates tradeoffs between sensitivity and specificity. The area under the curve is 0.78. The dashed, gray line indicates the performance expected by random chance.

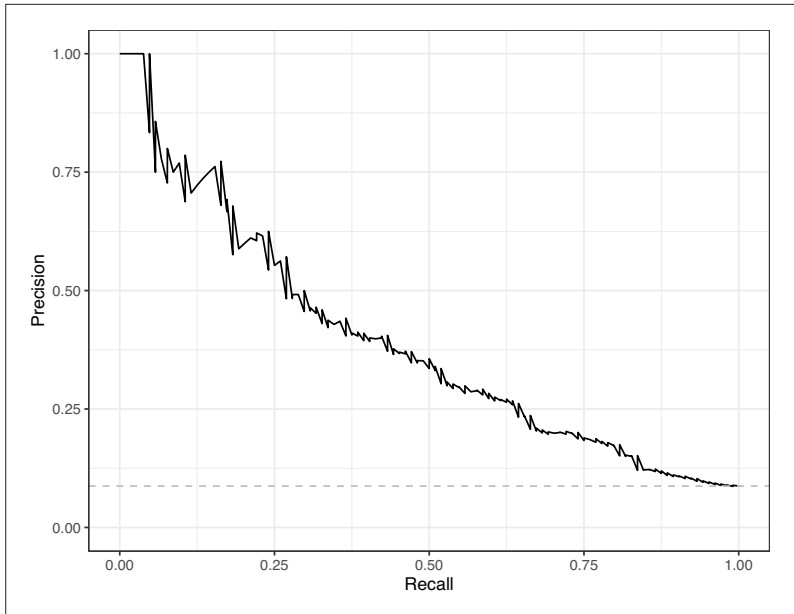

**Figure 21.** Precision-recall curve for the Convolutional Neural Network predictions for the images in the PubMed Central hold-out test set. This curve illustrates tradeoffs between precision and recall. The area under the curve is 0.39. The dashed, gray line indicates the frequency of the minority class ('Definitely problematic' images).

CVD simulations have facilitated the creation of colorblind-friendly palettes (*Olson and Brewer, 1997*), and they have led to algorithms that recolor images to become more accessible to people with CVD. Recolorization methods focus on enhancing color contrasts and preserving image natural-ness (*Zhu and Mao, 2021*). Many algorithms have been developed to compensate for dichromacy (*Jefferson and Harvey, 2007*; *Huang et al., 2007*; *Ruminski et al., 2010*; *Rasche et al., 2005*; *Machado and Oliveira, 2010*; *Ching and Sabudin, 2010*; *Ribeiro and Gomes, 2020*; *Li et al., 2020*; *Wang et al., 2021*; *Nakauchi and Onouchi, 2008*; *Zhu et al., 2019b*; *Ma et al., 2009*). These algo-rithms apply a variety of techniques including hue rotation, customized difference addition, node mapping, and generative adversarial networks (*Zhu and Mao, 2021*; *Li et al., 2020*). Many of these

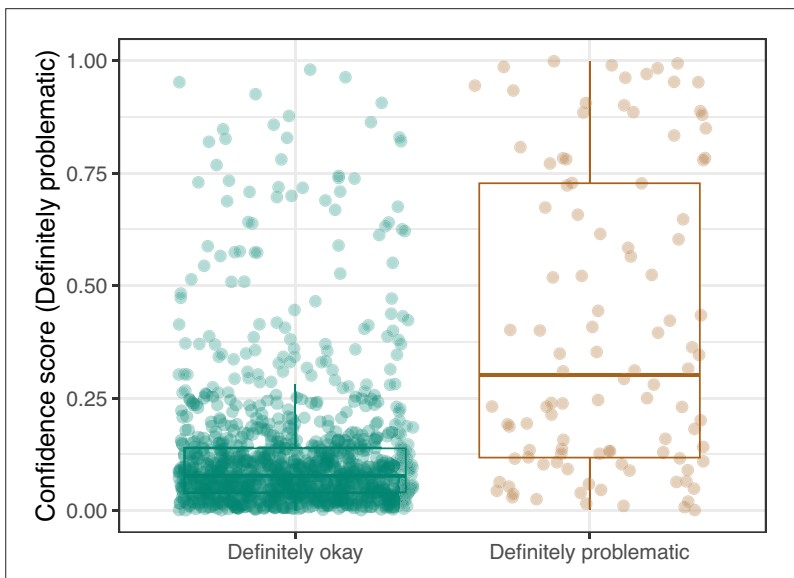

**Figure 22.** Convolutional Neural Network predictions for images in the PubMed Central hold-out test set. Each point represents the prediction for an individual image. Relatively high confidence scores indicate that the model had more confidence that a given image was 'Definitely problematic' for a person with deuteranopia.

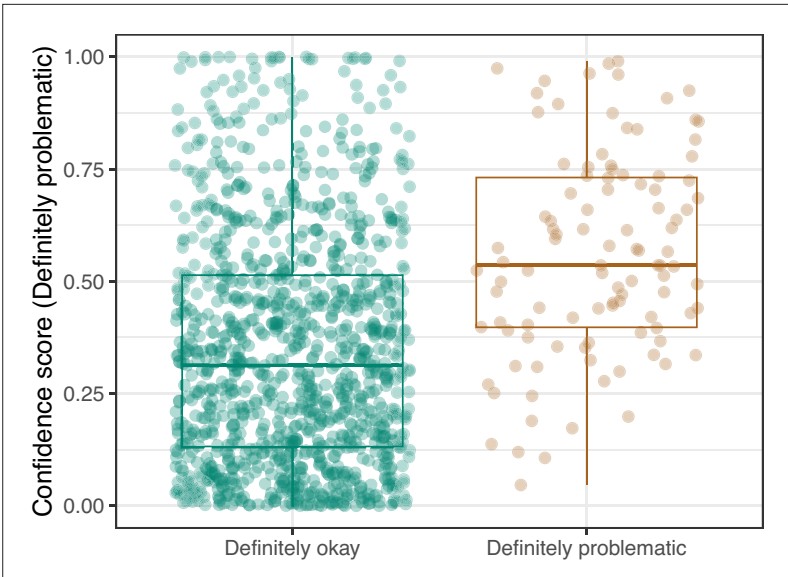

**Figure 23.** Logistic Regression predictions for the images in the PubMed Central hold-out test set. Each point represents the prediction for an individual image. Relatively high confidence scores indicate that the model had more confidence that a given image was 'Definitely problematic' for a person with deuteranopia.

methods have been tested for efficacy, both qualitatively and quantitatively (*Zhu and Mao, 2021*). Recolorization algorithms have been applied to PC displays, websites, and smart glasses (*Tanuwidjaja, 2014*). Despite the prevalence of these algorithms, current techniques have not been systematically compared and may sacrifice image naturalness to increase contrast. Additionally, recoloring may not improve the accessibility of some scientific figures because papers often reference colors in figure descriptions; recoloring the image could interfere with matching colors between the text and images.

An increase in available resources for making figures accessible to individuals with CVD has prompted some researchers to investigate whether these resources have been impactful in decreasing

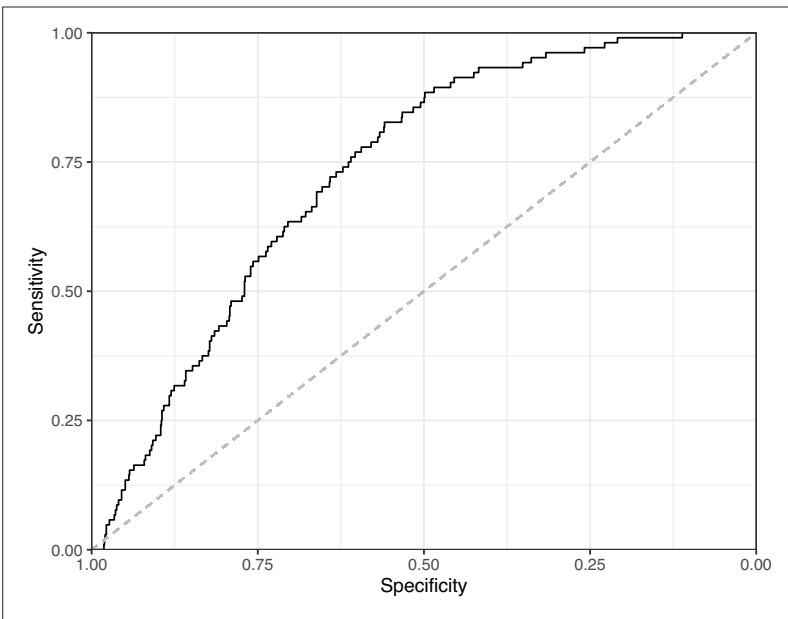

**Figure 24.** Receiver operating characteristic curve for the Logistic Regression predictions for the images in the PubMed Central hold-out test set. This curve illustrates tradeoffs between sensitivity and specificity. The area under the curve is 0.73. The dashed, gray line indicates the performance expected by random chance.

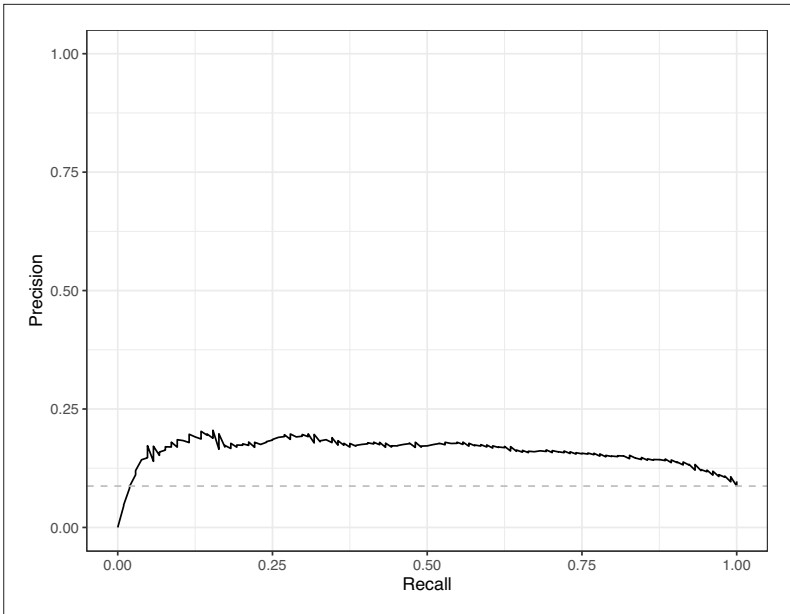

**Figure 25.** Precision-recall curve for the Logistic Regression predictions for the images in the PubMed Central hold-out test set. This curve illustrates tradeoffs between precision and recall. The area under the curve is 0.16. The dashed, gray line indicates the frequency of the minority class ('Definitely problematic' images).

the frequency of publishing scientific figures with problematic colors. Frane examined the prevalence of images in psychology journals that could be confusing to people with CVD (*Frane, 2015*). A group of panelists with CVD qualitatively evaluated 246 images and found that 13.8% of color figures caused difficulty for at least one panelist; this percentage is similar to our findings. They also found that in instructions to authors, journals rarely mentioned the importance of designing figures for CVD accessibility. Angerbauer et al. recruited crowdworkers to analyze a sample of 1,710 published images and to identify issues with the use of color (*Angerbauer et al., 2022*). On average, 60% of the sampled images were given a rating of 'accessible' across CVD types. From 2000 to 2019, they observed a slight increase in CVD accessibility for published figures.

Educational resources are available to researchers looking to make their figures suitable for people with CVD. For example, Jambor et al. provide guidelines and examples to help researchers avoid common problems (*Jambor et al., 2021*). *JetFighter* scans preprints from bioRxiv and searches for rainbow-based color schemes (*Saladi and Maggiolo, 2019*). When these are identified, *JetFighter* notifies the authors about page(s) that might need to be adjusted. However, as we have shown, the presence of particular color combinations does not necessarily indicate that an image is problematic to people with deuteranopia. Frequently, a more nuanced evaluation is necessary.

The *seaborn* Python package includes a 'colorblind' palette (*Waskom, 2021*). The *colorBlindness* package for R provides simulation tools and CVD-friendly palettes (*Ou, 2021*). The *scatterHatch* package facilitates creation of CVD-friendly scatter plots for single-cell data (*Guha et al., 2022*). When designing figures, researchers may find it useful to first design them so that key elements are distinguishable in grayscale. Then, color can be added—if necessary—to enhance the image. Color should not be used for the sole purpose of making an image aesthetically pleasing. Using minimal color avoids problems that arise from color pairing issues. Rainbow color maps, in particular, should be avoided. If a researcher finds it necessary to include problematic color pairings in figures, they can vary the saturation and intensity of the colors so they are more distinguishable to people with CVD. Many of the problematic figures that we identified in this study originated from fluorescence microscopy experiments, where red and green dyes were used. Choosing alternative color dyes could reduce this problem and improve the interpretability of microscopy images for people in all fields.

Our analysis has limitations. Firstly, it relied on deuteranopia simulations rather than the experiences of deuteranopes. However, by using simulations, the reviewers were capable of seeing two versions of each image: the original and a simulated version. We believe this is important in assessing

the extent to which deuteranopia could confound image interpretations. Conceivably, this could be done with deuteranopes after recoloration, but it is difficult to know whether deuteranopes would see the recolored images in the same way that non-deuteranopes see the original images. Secondly, because we used a single, relatively high severity threshold, our simulations do not represent the full spectrum of experiences that scientists with deuteranopia have. Thus, our findings and tools should be relevant to some (but not all) people with deuteranopia. Furthermore, recent evidence suggests that commonly used mathematical representations of color differences are unlikely to reflect human perceptions perfectly (*Bujack et al., 2022*). As methods evolve for more accurately simulating color perception, we will be more capable of estimating the extent to which scientific figures are problematic for deuteranopes. Thirdly, our evaluations focused on deuteranopia, the most common form of CVD. It will be important to address other forms of CVD, such as protanopia, in future work. Fourthly, we identified some images as 'Probably problematic' or 'Probably okay'. Using our review process, we were unable to draw firm conclusions about these images. To avoid adding noise to the classification analyses—we excluded these images and provided notes reflecting our reasoning. Future work may help to clarify these labels. Finally, our CNN model performed well at differentiating between 'Definitely okay' and 'Definitely problematic' images in the *eLife* hold-out test set; however, the model's predictive performance dropped considerably when applied to the PubMed Central hold-out test set. Many of the *eLife* images are from cell-related research, and we labeled many of these as problematic. Many other image types were also identified as unfriendly, including heat maps, line charts, maps, three-dimensional structural representations of proteins, photographs, network diagrams, etc. Our model may have developed a bias toward patterns specific to image types that are over-represented in *eLife*, affecting its performance for other journals. The PubMed Central Open Access Subset contains articles for thousands of journals, spanning diverse subdisciplines of biology and medicine. It seems likely that this diversity is a factor behind the drop in performance. Future efforts to review larger collections of PubMed Central articles could help to overcome this limitation.

By summarizing color patterns in more than 66,000 images and manually reviewing 8000 images, we have created an open data resource that other researchers can use to develop their own methods. Using all of these images, we trained a machine-learning model that predicts whether images are friendly to deuteranopes. It is available as a Web application (https://bioapps.byu.edu/colorblind_image_tester). Scientists and others can use it to obtain insights into whether individual images are accessible to scientists with deuteranopia. However, this tool should be used as a starting point only. Human judgment remains essential.

## Materials and methods
### Image acquisition
We evaluated images in articles from *eLife*, an open-access journal that publishes research in 'all areas of the life sciences and medicine'. Article content from this journal is released under a Creative Commons Attribution license. On June 1, 2022, we downloaded all available images from an Amazon Web Services storage bucket provided by journal staff. We also cloned a GitHub repository that *eLife* provides (https://github.com/elifesciences/elife-article-xml). This repository contains text and metadata from all articles published in the journal since its inception. For each article, we parsed the article identifier, digital object identifier, article type, article subject, and publication date. We excluded any article that was not published with the 'Research article' type. These articles were published between the years 2012 and 2022.

On March 21, 2024, we downloaded a list of articles from the *PMC Open Access Subset, 2003*. We filtered the articles to those published between 2012 and 2022 that used a CC BY license (https://creativecommons.org) and were categorized as research articles. This filtering resulted in 2,730,256 article candidates.

### Image summarization metrics
For each available image, we identified whether the image was either grayscale or contained colors. For each color image, we calculated a series of metrics to summarize the colors, contrasts, and distances between potentially problematic colors. These metrics have similarities to those used to assess recoloring algorithms, including global luminance error (*Kuhn et al., 2008*), local contrast error

(*Zhu et al., 2019a*), and global chromatic diversity (*Chen et al., 2011*; *Ma et al., 2009*). Before calculating the metrics, we sought to make the images more comparable to each other and to reduce the computational demands of analyzing the images. We scaled each image to a height of 300 pixels and generated a quantized version with a maximum of 256 colors. For each image, we then created a second version that simulated how a deuteranope would see the image. To facilitate these simulations, we used the *colorspace* package (*Stauffer et al., 2015*) and specified a 'severity' value of 0.8. Severity values range between 0 and 1 (with 1 being the most severe). We chose this threshold under the assumption that a mild severity level might not be stringent enough to identify a lack of contrast in the images. However, because many people with deuteranomaly do not have complete deuteranopia, this threshold reflects more moderate cases.

Our approach and rationale for these metrics are described below. In these descriptions, we refer to the quantized, resized images as 'original' images and their simulated counterparts as 'simulated' images.

- *Mean, pixel-wise color distance between the original and simulated image.* Our rationale was that the most problematic images would show relatively large overall differences between the original and simulated versions. When calculating these differences, we used version 2000 of Hunt's distance (*Hunt, 2005*), which quantifies red/green/blue (RGB) color differences in a three-dimensional space. This metric is symmetric, so the results are unaffected by the order in which the colors were specified; we used the absolute value of these distances.
- *Color-distance ratio between the original and simulated images for the color pair with the largest distance in the original image.* First, we excluded black, white, and gray colors. Second, we calculated the color distance (Hunt's method) between each unique pair of colors in the original image. Third, we calculated the color distance between the colors at the same locations in the simulated image. Fourth, we calculated the ratio between the original distance and the simulated distance. Our rationale was that problematic color pairs would have relatively high contrast (large distances) in the original images and relatively low contrast (small distances) in the simulated images. This approach is similar to that described by *Aisch, 2018*.
- *Number of color pairs that exhibited a high color-distance ratio between the original and simulated images.* This metric is similar to the previous one. However, instead of using the maximum ratio, we counted the number of color pairs with a ratio higher than five; this threshold was used by *Aisch, 2018*. Our rationale was that even if one color pair did not have an extremely high ratio, the presence of many high-ratio pairs would indicate a potential problem.
- *Proportion of pixels in the original image that used a color from one of the high-ratio color pairs.* Again, using a threshold of five, we identified unique colors among the color-distance pairs and counted the number of pixels in the original image that used any of these colors. Our rationale was that a relatively large number of pixels with potentially problematic colors may make an image as difficult for a deuteranope to interpret as an image with a few extremely low-contrast pixels.
- *Mean Euclidean distance between pixels for high-ratio color pairs.* First, we identified color pairs with a ratio higher than five. For each color pair, we identified pixels in the original image that used the two colors and calculated the Euclidean distance between those pixels in the image's two-dimensional layout. Then, we calculated the mean of these distances. Our rationale was that potentially problematic color pairs close together in an image would be more likely to cause problems than color pairs that are distant within the image.

After calculating these metrics for each available image, we calculated a ranked-based score. First, we assigned a rank to each image based on each of the metrics separately. For 'Mean Euclidean distance between pixels for high-ratio color pairs', relatively large values were given relatively high ranks (indicating that they were less problematic). For the other metrics, relatively small values were given relatively high ranks. Finally, we averaged the ranks to calculate a combined score for each image.

When analyzing images, calculating metrics, and creating figures and tables, we used the R statistical software (version 4.2.1) (*R Development Core Team, 2022*) and the following packages:

- colorspace (version 2.0–3) (*Stauffer et al., 2015*)
- diptest (0.77–1) (*Maechler, 2024*)
- doParallel (1.0.17) (*Corporation and Weston, 2022*)
- knitr (1.44) (*Xie, 2014*)
- magick (2.7.3) (*Ooms, 2021*) - interfaces with the ImageMagick software (6.9.10.23) (*Still, 2006*)

- pROC (1.17.0.1) (*Robin et al., 2011*)
- spacesXYZ (1.2–1) (*Davis, 2022*)
- tidyverse (2.0.0) (*Wickham et al., 2019*)
- xml2 (1.3.3) (*Wickham et al., 2021*)

## Qualitative image evaluation

We manually reviewed images to assess qualitatively whether visual characteristics were likely to be problematic for deuteranopes. Our intent was to establish a reference standard for evaluating the quantitative metrics we had calculated. Initially, we randomly sampled 1,000 *eLife* images from those we had downloaded. Two authors of this paper (HPS and AFO) reviewed each of the original (non-quantized, non-resized) images and the corresponding image that was simulated to reflect deuteranopia (severity = 0.8). Neither of these authors has been diagnosed with deuteranopia. This ensured the reviewers could compare the images with and without deuteranopia simulation. To avoid confirmation bias, neither author played a role in defining the quantitative metrics described above. Both authors reviewed the images and recorded observations based on four criteria:

1. Did an image contain shades of red, green, and/or orange that might be problematic for deuteranopes?
2. When an image contained potentially problematic color shades, did the color contrasts negate the potential problem? (The reviewers examined the images in their original and simulated forms when evaluating the contrasts).
3. When an image contained potentially problematic color shades, did within-image labels mitigate the potential problem?
4. When an image contained potentially problematic color shades, were the colors sufficiently, spatially distant from each other so that the colors were unlikely to be problematic?

After discussing a given image, the reviewers recorded a joint conclusion about whether the image was 'Definitely problematic', 'Probably problematic', 'Probably okay',, or 'Definitely okay'. For images that had no visually detectable color, the reviewers recorded 'Gray-scale'.

After this preliminary phase, we randomly selected an additional 4000 images from *eLife* and completed the same process. During the review process, we identified 36 cases where multiple versions of the same image had been sampled. We reviewed these versions manually and found that subsequent versions either had imperceptible differences or slight differences in the ways that sub-figures were laid out. None of these changes affected the colors used. Thus, we excluded the duplicate images and retained the earliest version of each image. The resulting 4964 images constituted a 'training set', which we used to evaluate our calculated metrics and to train classification models (see below).

Later, we randomly selected an additional 1000 images from *eLife*, which we used as a hold-out test set. Again, we excluded duplicate images and those for which different versions were present in the training set and hold-out test set. The same authors (HPS and AFO) performed the manual review process for these images.

From the candidate articles in the PubMed Central Open Access Subset, we randomly selected 2,000 articles. Two authors (HPS and AFO) manually reviewed these images.

## Classification analyses

We used classification algorithms to discriminate between images that we had manually labeled as either 'Definitely problematic' or 'Definitely okay'. Although it reduced our sample size, we excluded the 'Probably problematic' and 'Probably okay' images with the expectation that a smaller but more definitive set of examples would produce a more accurate model. Removing these images reduced our training set to 4501 images.

First, we evaluated our ability to classify images as 'Definitely problematic' or 'Definitely okay' based on the five metrics we devised. For this task, we used the following classification algorithms, which are designed for one-dimensional data: Random Forests (*Breiman, 2001*), k-nearest neighbors (*Fix and Hodges, 1989*), and logistic regression (*Nelder and Wedderburn, 1972*). We used implementations of these algorithms in scikit-learn (version 1.1.3) (*Pedregosa, 2011*) with default hyperparameters, other than two exceptions. We used the 'liblinear' solver for logistic regression, and we set the 'class_weight' hyperparameter to 'balanced' for Random Forests and Logistic Regression.

For evaluation, we used three iterations of five-fold cross validation; we used multiple iterations to account for variability in model performance and to ensure reliable estimates across different subsets of the data. For the test samples in each fold, we calculated the area under the receiver operating characteristic curve (AUROC) (*Tanner and Swets, 2001*; *Swets, 1988*) using the *yardstick* package (1.2.0) (*Kuhn et al., 2023*); we calculated the area under the precision-recall curve (AUPRC) using the *PRROC* package (1.3.1) (*Grau et al., 2015*). We calculated the median AUROC and AUPRC across the folds and then averaged them across the three iterations.

Second, we evaluated our ability to classify the images as 'Definitely problematic' or 'Definitely okay' based on the images themselves. We used a CNN because CNN models are capable of handling two-dimensional inputs and accounting for spatial patterns and colors within images. To generate the CNN models, we used the Tensorflow (2.10.0) and Keras (2.10.0) packages (*Abadi, 2016*; *Géron, 2022*). To support transfer learning (described below), we scaled both dimensions of each image to 224. To select from different configurations, we again used three iterations of fivefold cross-validation (with the same assignments as the earlier classification analysis). Each model configuration extended a baseline configuration that had eight, two-dimensional, convolutional layers; each layer used batch normalization and the *ReLU* activation function. Subsequent layers increased in size, starting with 32 nodes and increasing to 64, 128, 256, 512, and 728. We trained for 30 epochs with an Adam optimization set, a learning rate of 1e-3, and the binary cross-entropy loss function. The output layer used a sigmoid activation function.

In addition to the baseline configuration, we tested 22 model configurations based on combinations of the following techniques:

- *Class weighting* - To address class imbalance (most images were 'Definitely okay' in the training set), we increased the weight of the minority class ('Definitely problematic') proportionally to its frequency in the training set.
- *Early stopping* - During model training, classification performance on the (internal) validation set is monitored to identify an epoch when the performance is no longer improving or has begun to degrade; the goal of this technique is to find a balance between underfitting and overfitting.
- *Random flipping and rotation* - In an attempt to prevent overfitting, we enabled random, horizontal flipping of training images and data augmentation via differing amounts of random image rotation (*Wong et al., 2016*). We evaluated rotation thresholds of 0.2 and 0.3.
- *Dropout* - Again to prevent overfitting, we temporarily removed a subset of neurons from the network. We evaluated dropout rates of 0.2 and 0.5.
- *Transfer learning* - This technique uses a corpus of ancillary images such that model building is informed by patterns observed previously. We evaluated two corpuses: MobileNetV2 (*Sandler et al., 2019*; *Krizhevsky et al., 2012*) and ResNet50 (*He et al., 2016*). MobileNetV2 is a 53-layer convolutional neural network trained on more than a million images from the ImageNet database to classify images with objects into 1000 categories. ResNet50 is a 50-layer convolutional neural network, similarly trained. MobileNetV2 is designed for use on mobile devices, so it is optimized to be lightweight. As such, MobileNetV2 uses 3.4 million parameters, while ResNet50 uses over 25 million trainable parameters. When we applied transfer learning from either ResNet50 or MobileNetV2, we did not use the baseline configuration. Instead, we added a global pooling function into a dense layer. Because our training dataset was relatively small, we expected that adding fewer layers might reduce the risk of overfitting.
- *Fine tuning* - In combination with transfer learning, we sometimes employed a two-phase training process involving an initial training phase and a fine-tuning phase. In the initial phase, we used a pre-trained model (ResNet50 or MobileNetV2) as the base model. During this phase, we froze the layers of the base model to retain the learned weights and trained the combined model for 30 epochs with a learning rate of 0.001. In the fine-tuning phase, we unfroze the layers of the base model to allow the entire model to be retrained. To avoid large adjustments that could disrupt the pre-trained weights, we reduced the learning rate to 1e-5 and trained the model for an additional 15 epochs. This phase enabled the model to make subtle updates to the pre-trained weights. After fine-tuning, the layers of the base model were refrozen.

When training each model configuration, we used AUROC to evaluate the predictive performance on the internal validation sets. After comparing the model configurations via cross validation, we used the full training set to train a model, which we used to make predictions for the hold-out test sets. We calculated the following: true positives, false positives, true negatives, false negatives, accuracy, precision (positive predictive value), recall (sensitivity), AUROC, and AUPRC.

## Web application

We created a Web application using the *Node.js* framework (*OpenJS Foundation, 2024*). The application enables researchers to evaluate uploaded images. First, users upload an image in PNG or JPEG format. The application displays the image alongside a deuteranopia-simulated version of the image. For simulation, we implemented the *Machado et al., 2009* matrix for deuteranopia in Javascript with a 'severity' value of 0.8, the same parameter used in training. If the user requests it, the application predicts whether the image is likely to be problematic for a deuteranope; the prediction includes a probabilistic score so that users can assess the model's confidence level. With the intent to maximize the generalizability of these predictions, we trained a model using images from the training set and both holdout test sets. To facilitate execution of the CNN within the Web application, we used Tensorflow.js (version 4.0.0) (*TensorFlow, 2019*).

## Additional information

### Funding
No external funding was received for this work.

### Author contributions
Harlan P Stevens, Conceptualization, Data curation, Software, Formal analysis, Validation, Investigation, Methodology, Writing – original draft, Writing – review and editing; Carly V Winegar, Conceptualization, Software, Investigation, Methodology, Writing – original draft, Writing – review and editing; Arwen F Oakley, Data curation, Formal analysis, Writing – review and editing; Stephen R Piccolo, Conceptualization, Resources, Software, Formal analysis, Supervision, Investigation, Visualization, Methodology, Writing – original draft, Project administration, Writing – review and editing

### Author ORCIDs
Stephen R Piccolo ⓘ https://orcid.org/0000-0003-2001-5640

Reviewer #1 (Public Review): https://doi.org/10.7554/eLife.95524.3.sa1
Reviewer #2 (Public Review): https://doi.org/10.7554/eLife.95524.3.sa2
Reviewer #3 (Public Review): https://doi.org/10.7554/eLife.95524.3.sa3
Author response https://doi.org/10.7554/eLife.95524.3.sa4

## Additional files

### Supplementary files
• Supplementary file 1. Supplementary tables. (A) Predictive performance for metrics that characterize potentially problematic aspects of images. We calculated five metrics, as well as a rank-based, combined score and assessed their ability to categorize images as "Definitely okay" or "Definitely problematic". (B) Predictive performance for classification algorithms that used the image metrics as inputs. We used classification algorithms to categorize images as "Definitely okay" or "Definitely problematic". These results indicate the algorithms' performance after cross validation on the training set. (C) Predictive performance for Convolutional Neural Network models that used the images as inputs. We tested 23 model configurations via cross validation on the training set, evaluating each model's ability to categorize images as "Definitely okay" or "Definitely problematic".

• Supplementary file 2. Results of manual curation for 5,000 images from the training set.

• Supplementary file 3. Results of manual curation for 1,000 images from the *eLife* hold-out test set.

• Supplementary file 4. Performance metrics for predictions made on hold-out test sets. This file provides a variety of classification metrics for the two hold-out test sets. AUROC = area under receiver operating characteristic curve. AUPRC = area under precision-recall curve.

• Supplementary file 5. Results of manual review of misclassified images from the hold-out test set. We manually reviewed each image that we had previously classified as "Definitely okay" but that the model predicted as "Definitely problematic" (or vice versa). The Conclusion column indicates

our categorical reevaluation of each image. "Unclear"=We continue to conclude that our manual label was correct, and it is unclear what confused the model. "Understandable"=We continue to conclude that our manual label was correct, and we think we understand what confused the model. "Agree"=We acknowledge that the manual label was incorrect, and the model helped us identify that.

• Supplementary file 6. Results of manual curation for 2,000 images used from the PubMed Central hold-out test set.

• MDAR checklist

### Data availability

The images we used for evaluation and the trained TensorFlow models are stored in an Open Science Framework repository (https://osf.io/8yrkb). It also includes folders with images marked as "friendly" or "unfriendly" to facilitate examination of images in either category. The code for processing and analyzing the images is available at GitHub (copy archived at *Piccolo and Stevens, 2024*). That repository also includes the calculated metrics, cross-validation assignments, results of image curation, and outputs of the classification algorithms. The Web application code is available at GitHub (copy archived at *Piccolo, 2024*). The code for processing and analyzing the images is available as a Zenodo archive (https://doi.org/10.5281/zenodo.13366997). It includes the calculated metrics, cross-validation assignments, results of image curation, and outputs of the classification algorithms. The Web application code is available as a Zenodo archive (https://doi.org/10.5281/zenodo.13367011).

The following dataset was generated:

| Author(s) | Year | Dataset title | Dataset URL | Database and Identifier |
|---|---|---|---|---|
| Piccolo SR | 2023 | Supplementary images from eLife articles and the PubMed Open Access Data Subset | https://doi.org/10.17605/OSF.IO/8YRKB | Open Science Framework, 10.17605/OSF.IO/8YRKB |

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
