## [Editor Report · eLife assessment]

In this **important** study, the authors manually assessed randomly selected images published in eLife between 2012 and 2022 to determine whether they were accessible for readers with deuteranopia, the most common form of color vision deficiency. They then developed an automated tool designed to classify figures and images as either "friendly" or "unfriendly" for people with deuteranopia. Such a tool could be used by journals or researchers to monitor the accessibility of figures and images, and the evidence for its utility was **solid**: it performed well for eLife articles, but performance was weaker for a broader dataset of PubMed articles, which were not included in the training data. The authors also provide code that readers can download and run to test their own images, and this may be of most use for testing the tool, as there are already several free, user-friendly recoloring programs that allow users to see how images would look to a person with different forms of color vision deficiency. Automated classifications are of most use for assessing many images, when the user does not have the time or resources to assess each image individually.

---

## [Referee Report · Reviewer #1 (Public Review)]

The authors of this study developed a software application, which aims to identify images as either "friendly" or "unfriendly" for readers with deuteranopia, the most common color-vision deficiency. Using previously published algorithms that recolor images to approximate how they would appear to a deuteranope (someone with deuteranopia), authors first manually assessed a set of images from biology-oriented research articles published in eLife between 2012 and 2022, as well as an additional hold-out set of 2000 articles selected randomly from the PubMed Central Open Access Subset. The researchers identified 636 out of 4964 images as difficult to interpret ("unfriendly") for deuteranopes in the eLife dataset. In the PubMed Central dataset 104 out of 1191 non-grayscale images were identified as unfriendly. The results for the eLife dataset show a decrease in "unfriendly" images over time and a higher probability for articles from cell-oriented research fields to contain "unfriendly" images.

The researchers used the manually classified images from eLife to develop, train, and validate an automated screening tool. They also created a user-friendly web application of the tool, where users can upload images and be informed about the status of each image as "friendly" or "unfriendly" for deuteranopes.

Strengths:

The authors have identified an important accessibility issue in the scientific literature: the use of color combinations that make figures difficult to interpret for people with color-vision deficiency. The metrics proposed and evaluated in the study are a valuable theoretical contribution. The automated screening tool they provide is well-documented, open source, and relatively easy to install and use. It has the potential to provide a useful service to the scientists who want to make their figures more accessible. The data are open and freely accessible, well documented, and a valuable resource for further research. The manuscript is well-written, logically structured, and easy to follow.

Weaknesses:

(1) The authors themselves acknowledge the limitations that arise from the way they defined what constitutes an "unfriendly" image. There is a missed chance here to have engaged deuteranopes as stakeholders earlier in the experimental design. This would have allowed to determine to what extent spatial separation and labelling of problematic color combinations responds to their needs and whether setting the bar at a simulated severity of 80% is inclusive enough. A slightly lowered barrier is still a barrier to accessibility.

(2) The use of training images from a single journal limits the generalizability of the empirical findings as well as of the automated screening tool itself. This is evidenced by a decrease in performance of the tool on the holdout dataset from PubMed Central. Machine-learning algorithms are highly configurable but also notorious for their lack of transparency and for being easily biased by the training data set. A quick and unsystematic test of the web application shows that the classifier works well for electron microscopy images but fails at recognizing the classical diagnostic images for color-vision deficiency (Ishihara test images) as "unfriendly". A future iteration of the tool should be trained on a wider variety of images, ideally enriched with diagnostic images found in scientific publications.

---

## [Referee Report · Reviewer #2 (Public Review)]

Summary:

An analysis of images in the biology literature that are problematic for people with a color-vision deficiency (CVD) is presented, along with a machine learning-based model trained on an eLife dataset to identify such images and a web application that uses the model to flag problematic images. Their analysis reveals that about 13% of the images could be problematic for people with CVD and that the frequency of such images decreased over time. Their best model (convolutional neural network, CNN) yields 0.89 AUROC score and 0.77 AUPRC on held-out eLife articles, but lower scores (0.78 and 0.39, respectively). It is proposed that their approach could help making biology literature accessible to diverse audiences.

Strengths:

The manuscript focuses on an important yet mostly overlooked problem and makes contributions both in expanding our understanding of the extent of the problem and in developing solutions to mitigate the problem. The paper is generally well-written and clearly organized. Their CVD simulation combines five different metrics. The dataset has been assessed by two researchers and is likely to be of high-quality. Machine learning algorithm used (CNN) is an appropriate choice for the problem. The evaluation of various hyperparameters for the CNN model is extensive.

Weaknesses:

While the study has significant strengths, it also has some limitations. Specifically, the focus on one type of CVD (deuteranopia) and selecting images from a single journal (eLife) for training limit the generalizability of the models. This is, to some extent, shown by applying the model to PMC articles, which yields lower performance. "Probably problematic" and "probably okay" classes are excluded from the analysis.

---

## [Referee Report · Reviewer #3 (Public Review)]

Summary:

This work focuses on accessibility of scientific images for individuals with color vision deficiencies, particularly deuteranopia. The research involved an analysis of images from eLife and PubMed Central published in 2012-2022. The authors manually reviewed nearly 7,000 images, comparing them with simulated versions representing the perspective of individuals with deuteranopia, and also evaluated several methods to automatically detect such images including training a machine-learning algorithm to do so, which performed the best. The authors found that nearly 13% of the images could be challenging for people with deuteranopia to interpret. There was a trend toward a decrease in problematic images over time, which is encouraging.

After the first round of review, the addition of a random sample of biomedical articles in the evaluation set strengthens the generalizability of the algorithm, and the change to evaluate articles at the article level to address pseudoreplication is appropriate.

---

## [Author Response]

The following is the authors’ response to the original reviews.

**eLife assessment**
In this important study, the authors manually assessed randomly selected images published in eLife between 2012 and 2020 to determine whether they were accessible for readers with deuteranopia, the most common form of color vision deficiency. They then developed an automated tool designed to classify figures and images as either "friendly" or "unfriendly" for people with deuteranopia. While such a tool could be used by publishers, editors or researchers to monitor accessibility in the research literature, the evidence supporting the tools' utility was incomplete. The tool would benefit from training on an expanded dataset that includes different image and figure types from many journals, and using more rigorous approaches when training the tool and assessing performance. The authors also provide code that readers can download and run to test their own images. This may be of most use for testing the tool, as there are already several free, user-friendly recoloring programs that allow users to see how images would look to a person with different forms of color vision deficiency. Automated classifications are of most use for assessing many images, when the user does not have the time or resources to assess each image individually.

Thank you for this assessment. We have responded to the comments and suggestions in detail below. One minor correction to the above statement: the randomly selected images published in eLife were from articles published between 2012 and 2022 (not 2020).

**Public Reviews:**

**Reviewer #1 (Public Review):**
Summary:The authors of this study developed a software application, which aims to identify images as either "friendly" or "unfriendly" for readers with deuteranopia, the most common color-vision deficiency. Using previously published algorithms that recolor images to approximate how they would appear to a deuteranope (someone with deuteranopia), authors first manually assessed a set of images from biology-oriented research articles published in eLife between 2012 and 2022. The researchers identified 636 out of 4964 images as difficult to interpret ("unfriendly") for deuteranopes. They claim that there was a decrease in "unfriendly" images over time and that articles from cell-oriented research fields were most likely to contain "unfriendly" images. The researchers used the manually classified images to develop, train, and validate an automated screening tool. They also created a user-friendly web application of the tool, where users can upload images and be informed about the status of each image as "friendly" or "unfriendly" for deuteranopes.Strengths:The authors have identified an important accessibility issue in the scientific literature: the use of color combinations that make figures difficult to interpret for people with color-vision deficiency. The metrics proposed and evaluated in the study are a valuable theoretical contribution. The automated screening tool they provide is well-documented, open source, and relatively easy to install and use. It has the potential to provide a useful service to the scientists who want to make their figures more accessible. The data are open and freely accessible, well documented, and a valuable resource for further research. The manuscript is well written, logically structured, and easy to follow.

We thank the reviewer for these comments.

Weaknesses:(1) The authors themselves acknowledge the limitations that arise from the way they defined what constitutes an "unfriendly" image. There is a missed chance here to have engaged deuteranopes as stakeholders earlier in the experimental design. This would have allowed [them] to determine to what extent spatial separation and labelling of problematic color combinations responds to their needs and whether setting the bar at a simulated severity of 80% is inclusive enough. A slightly lowered barrier is still a barrier to accessibility.

We agree with this point in principle. However, different people experience deuteranopia in different ways, so it would require a large effort to characterize these differences and provide empirical evidence about many individuals' interpretations of problematic images in the "real world." In this study, we aimed to establish a starting point that would emphasize the need for greater accessibility, and we have provided tools to begin accomplishing that. We erred on the side of simulating relatively high severity (but not complete deuteranopia). Thus, our findings and tools should be relevant to some (but not all) people with deuteranopia. Furthermore, as noted in the paper, an advantage of our approach is that "by using simulations, the reviewers were capable of seeing two versions of each image: the original and a simulated version." We believe this step is important in assessing the extent to which deuteranopia could confound image interpretations. Conceivably, this could be done with deuteranopes after recoloration, but it is difficult to know whether deuteranopes would see the recolored images in the same way that non-deuteranopes see the original images. It is also true that images simulating deuteranopia may not perfectly reflect how deuteranopes see those images. It is a tradeoff either way. We have added comments along these lines to the paper.

(2) The use of images from a single journal strongly limits the generalizability of the empirical findings as well as of the automated screening tool itself. Machine-learning algorithms are highly configurable but also notorious for their lack of transparency and for being easily biased by the training data set. A quick and unsystematic test of the web application shows that the classifier works well for electron microscopy images but fails at recognizing red-green scatter plots and even the classical diagnostic images for color-vision deficiency (Ishihara test images) as "unfriendly". A future iteration of the tool should be trained on a wider variety of images from different journals.

Thank you for these comments. We have reviewed an additional 2,000 images, which were randomly selected from PubMed Central. We used our original model to make predictions for those images. The corresponding results are now included in the paper.

We agree that many of the images identified as being "unfriendly" are microscope images, which often use red and green dyes. However, many other image types were identified as unfriendly, including heat maps, line charts, maps, three-dimensional structural representations of proteins, photographs, network diagrams, etc. We have uploaded these figures to our Open Science Framework repository so it's easier for readers to review these examples. We have added a comment along these lines to the paper.

The reviewer mentioned uploading red/green scatter plots and Ishihara test images to our Web application and that it reported they were friendly. Firstly, it depends on the scatter plot. Even though some such plots include green and red, the image's scientific meaning may be clear. Secondly, although the Ishihara images were created as informal tests for humans, these images (and ones similar to them) are not in eLife journal articles (to our knowledge) and thus are not included in our training set. Thus, it is unsurprising that our machine-learning models would not classify such images correctly as unfriendly.

(3) Focusing the statistical analyses on individual images rather than articles (e.g. in figures 1 and 2) leads to pseudoreplication. Multiple images from the same article should not be treated as statistically independent measures, because they are produced by the same authors. A simple alternative is to instead use articles as the unit of analysis and score an article as "unfriendly" when it contains at least one "unfriendly" image. In addition, collapsing the counts of "unfriendly" images to proportions loses important information about the sample size. For example, the current analysis presented in Fig. 1 gives undue weight to the three images from 2012, two of which came from the same article. If we perform a logistic regression on articles coded as "friendly" and "unfriendly" (rather than the reported linear regression on the proportion of "unfriendly" images), there is still evidence for a decrease in the frequency of "unfriendly" eLife articles over time.

Thank you for taking the time to provide these careful insights. We have adjusted these statistical analyses to focus on articles rather than individual images. For Figure 1, we treat an article as "Definitely problematic" if any image in the article was categorized as "Definitely problematic." Additionally, we no longer collapse the counts to proportions, and we use logistic regression to summarize the trend over time. The overall conclusions remain the same.

Another issue concerns the large number of articles (>40%) that are classified as belonging to two subdisciplines, which further compounds the image pseudoreplication. Two alternatives are to either group articles with two subdisciplines into a "multidisciplinary" group or recode them to include both disciplines in the category name.

Thank you for this insight. We have modified Figure 2 so that it puts all articles that have been assigned two subdisciplines into a "Multidisciplinary" category. The overall conclusions remain the same.

(4) The low frequency of "unfriendly" images in the data (under 15%) calls for a different performance measure than the AUROC used by the authors. In such imbalanced classification cases the recommended performance measure is precision-recall area under the curve (PR AUC: https://doi.org/10.1371%2Fjournal.pone.0118432) that gives more weight to the classification of the rare class ("unfriendly" images).

We now calculate the area under the precision-recall curve and provide these numbers (and figures) alongside the AUROC values (and figures). We agree that these numbers are informative; both metrics lead to the same overall conclusions.

**Reviewer #2 (Public Review):**
Summary:An analysis of images in the biology literature that are problematic for people with a color-vision deficiency (CVD) is presented, along with a machine learning-based model to identify such images and a web application that uses the model to flag problematic images. Their analysis reveals that about 13% of the images could be problematic for people with CVD and that the frequency of such images decreased over time. Their model yields 0.89 AUC score. It is proposed that their approach could help making biology literature accessible to diverse audiences.Strengths:The manuscript focuses on an important yet mostly overlooked problem, and makes contributions both in expanding our understanding of the extent of the problem and in developing solutions to mitigate the problem. The paper is generally well-written and clearly organized. Their CVD simulation combines five different metrics. The dataset has been assessed by two researchers and is likely to be of high-quality. Machine learning algorithm used (convolutional neural network, CNN) is an appropriate choice for the problem. The evaluation of various hyperparameters for the CNN model is extensive.

We thank the reviewer for these comments.

Weaknesses:The focus seems to be on one type of CVD (deuteranopia) and it is unclear whether this would generalize to other types.

We agree that it would be interesting to perform similar analyses for protanopia and other color-vision deficiencies. But we leave that work for future studies.

The dataset consists of images from eLife articles. While this is a reasonable starting point, whether this can generalize to other biology/biomedical articles is not assessed.

This is an important point. We have reviewed an additional 2,000 images, which were randomly selected from PubMed Central, and used our original model to make predictions for those images. The corresponding results are now included in the paper.

"Probably problematic" and "probably okay" classes are excluded from the analysis and classification, and the effect of this exclusion is not discussed.

We now address this in the Discussion section.

Machine learning aspects can be explained better, in a more standard way.

Thank you. We address this comment in our responses to your comments below.

The evaluation metrics used for validating the machine learning models seem lacking (e.g., precision, recall, F1 are not reported).

We now provide these metrics (in a supplementary file).

The web application is not discussed in any depth.

The paper includes a paragraph about how the Web application works and which technologies we used to create it. We are unsure which additional aspects should be addressed.

**Reviewer #3 (Public Review):**
Summary:This work focuses on accessibility of scientific images for individuals with color vision deficiencies, particularly deuteranopia. The research involved an analysis of images from eLife published in 2012-2022. The authors manually reviewed nearly 5,000 images, comparing them with simulated versions representing the perspective of individuals with deuteranopia, and also evaluated several methods to automatically detect such images including training a machine-learning algorithm to do so, which performed the best. The authors found that nearly 13% of the images could be challenging for people with deuteranopia to interpret. There was a trend toward a decrease in problematic images over time, which is encouraging.Strengths:The manuscript is well organized and written. It addresses inclusivity and accessibility in scientific communication, and reinforces that there is a problem and that in part technological solutions have potential to assist with this problem.The number of manually assessed images for evaluation and training an algorithm is, to my knowledge, much larger than any existing survey. This is a valuable open source dataset beyond the work herein.The sequential steps used to classify articles follow best practices for evaluation and training sets.

We thank the reviewer for these comments.

Weaknesses:I do not see any major issues with the methods. The authors were transparent with the limitations (the need to rely on simulations instead of what deuteranopes see), only capturing a subset of issues related to color vision deficiency, and the focus on one journal that may not be representative of images in other journals and disciplines.

We thank the reviewer for these comments. Regarding the last point, we have reviewed an additional 2,000 images, which were randomly selected from PubMed Central, and used our original model to make predictions for those images. The corresponding results are now included in the paper.

**Recommendations for the authors:**

**Reviewer #1 (Recommendations For The Authors):**
N/A

Thank you.

**Reviewer #2 (Recommendations For The Authors):**
- The web application link can be provided in the Abstract for more visibility.

We have added the URL to the Abstract.

- They focus on deuteranopia in this paper. It seems that protanopia is not considered. Why? What are the challenges in considered this type of CVD?

We agree that it would be interesting to perform similar analyses for protanopia and other color-vision deficiencies. But we leave that work for future studies. Deuteranopia is the most common color-vision deficiency, so we focused on the needs of these individuals as a starting point.

- The dataset is limited to eLife articles. More discussion of this limitation is needed. Couldn't one also include some papers from PMC open access dataset for comparison?

We have reviewed an additional 2,000 images, which we randomly selected from PubMed Central, and used our original model to make predictions for those images. The corresponding results are now included in the paper.

- An analysis of the effect of selecting a severity value of 0.8 can be included.

We agree that this would be interesting, but we leave it for future work.

- "Probably problematic" and "probably okay" classes are excluded from analysis, which may oversimplify the findings and bias the models. It would have been interesting to study these classes as well.

We agree that this would be interesting, but we leave it for future work. However, we have added a comment to the Discussion on this point.

- Some machine learning aspects are discussed in a non-standard way. Class weighting or transfer learning would not typically be considered hyperparameters."corpus" is not a model. Description of how fine-tuning was performed could be clearer.

We have updated this wording to use more appropriate terminology to describe these different "configurations." Additionally, we expanded and clarified our description of fine tuning.

- Reporting performance on the training set is not very meaningful. Although I understand this is cross-validated, it is unclear what is gained by reporting two results. Maybe there should be more discussion of the difference.

We used cross validation to compare different machine-learning models and configurations. Providing performance metrics helps to illustrate how we arrived at the final configurations that we used. We have updated the manuscript to clarify this point.

- True positives, false positives, etc. are described as evaluation metrics. Typically, one would think of these as numbers that are used to calculate evaluation metrics, like precision (PPV), recall (sensitivity), etc. Furthermore, they say they measure precision, recall, precision-recall curves, but I don't see these reported in the manuscript. They should be (especially precision, recall, F1).

We have clarified this wording in the manuscript.

- There are many figures in the supplementary material, but not much interpretation/insights provided. What should we learn from these figures?

We have revised the captions and now provide more explanations about these figures in the manuscript.

- CVD simulations are mentioned (line 312). It is unclear whether these methods could be used for this work and if so, why they were not used. How do the simulations in this work compare to other simulations?

This part of the manuscript refers to recolorization techniques, which attempt to make images more friendly to people with color vision deficiencies. For our paper, we used a form of recolorization that simulates how a deuteranope would see a figure in its original form. Therefore, unless we misunderstand the reviewer's question, these two types of simulation have distinct purposes and thus are not comparable.

- relu -> ReLU

We have corrected this.

**Reviewer #3 (Recommendations For The Authors):**
The title can be more specific to denote that the survey was done in eLife papers in the years 2012-2022. Similarly, this should be clear in the abstract instead of only "images published in biology-oriented research articles".

Thank you for this suggestion. Because we have expanded this work to include images from PubMed Central papers, we believe the title is acceptable as it stands. We updated the abstract to say, "images published in biology- and medicine-oriented research articles"

Two mentions of existing work that I did not see are to Jambor and colleagues' assessment on color accessibility in several fields: https://www.ncbi.nlm.nih.gov/pmc/articles/PMC8041175/, and whether this work overlaps with the 'JetFighter' tool(https://elifesciences.org/labs/c2292989/jetfighter-towards-figure-accuracy-and-accessibility).

Thank you for bringing these to our attention. We have added a citation to Jambor, et al.

We also mention JetFighter and describe its uses.

Similarly, on Line 301: Significant prior work has been done to address and improve accessibility for individuals with CVD. This work can be generally categorized into three types of studies: simulation methods, recolorization methods, and estimating the frequency of accessible images.- One might mention education as prior work as well, which might in part be contributing to a decrease in problematic images (e.g., https://www.ncbi.nlm.nih.gov/pmc/articles/PMC8041175/)

We now suggest that there are four categories and include education as one of these.

Line 361, when discussing resources to make figures suitable, the authors may consider citing this paper about an R package for single-cell data: https://elifesciences.org/articles/82128

Thank you. We now cite this paper.

The web application is a good demonstration of how this can be applied, and all code is open so others can apply the CNN in their own uses cases. Still, by itself, it is tedious to upload individual image files to screen them. Future work can implement this into a workflow more typical to researchers, but I understand that this will take additional resources beyond the scope of this project. The demonstration that these algorithms can be run with minimal resources in the browser with tensorflow.js is novel.

Thank you.

General:It is encouraging that 'definitely problematic' images have been decreasing over time in eLife. Might this have to do with eLife policies? I could not quickly find if eLife has checks in place for this, but given that JetFighter was developed in association with eLife, I wonder if there is an enhanced awareness of this issue here vs. other journals.

This is possible. We are not aware of a way to test this formally.